# WEAKLY-SUPERVISED KNOWLEDGE GRAPH ALIGNMENT WITH ADVERSARIAL LEARNING

## ABSTRACT

Aligning knowledge graphs from different sources or languages, which aims to align both the entity and relation, is critical to a variety of applications such as knowledge graph construction and question answering. Existing methods of knowledge graph alignment usually rely on a large number of aligned knowledge triplets to train effective models. However, these aligned triplets may not be available or are expensive to obtain for many domains. Therefore, in this paper we study how to design fully-unsupervised methods or weakly-supervised methods, i.e., to align knowledge graphs without or with only a few aligned triplets. We propose an unsupervised framework based on adversarial training, which is able to map the entities and relations in a source knowledge graph to those in a target knowledge graph. This framework can be further seamlessly integrated with existing supervised methods, where only a limited number of aligned triplets are utilized as guidance. Experiments on real-world datasets prove the effectiveness of our proposed approach in both the weakly-supervised and unsupervised settings.

## 1 INTRODUCTION

Knowledge graphs represent a collection of knowledge facts and are quite popular in the real world. Each fact is represented as a triplet $(h, r, t)$, meaning that the head entity $h$ has the relation $r$ with the tail entity $t$. Examples of real-world knowledge graphs include instances which contain knowledge facts from general domain in different languages (Freebase [1], DBPedia (Auer et al., 2007), Yago (Suchanek et al., 2007), WordNet [2]) or facts from specific domains such as biomedical ontology (UMLS [3]). Knowledge graphs are critical to a variety of applications such as question answering (Bordes et al. (2014)) and semantic search (Guha et al. (2003)), which are attracting growing interest recently in both academia and industry communities.

In practice, each knowledge graph is usually constructed from a single source or language, the coverage of which is limited. To enlarge the coverage and construct more unified knowledge graphs, a natural idea is to integrate multiple knowledge graphs from different sources or languages (Arens et al. (1993)). However, different knowledge graphs use distinct symbol systems to represent entities and relations, which are not compatible. As a result, it is necessary to align entities and relations across different knowledge graphs (a.k.a., knowledge graph alignment) before integrating them.

Indeed, there are some recent studies focusing on aligning entities and relations from a source knowledge graph to a target knowledge graph ((Zhu et al., 2017a); (Chen et al., 2017a); (Chen et al., 2017b)). These methods typically represent entities and relations in a low-dimensional space, and meanwhile learn a mapping function to align entities and relations from the source knowledge graph to the target one. However, these methods usually rely on a large number of aligned triplets as labeled data to train effective alignment models. In reality, the aligned triplets may not be available or can be expensive to obtain, making existing methods fail to achieve satisfactory results. Therefore, we are seeking for an unsupervised or weakly-supervised approach, which is able to align knowledge graphs with a few or even without labeled data.

---

[1] https://developers.google.com/freebase/
[2] https://wordnet.princeton.edu/
[3] https://www.nlm.nih.gov/research/umls/

In this paper, we propose an unsupervised approach for knowledge graph alignment with the adversarial training framework Goodfellow et al. (2014). Our proposed approach aims to learn alignment functions, i.e., $P_e(e^{tgt}|e^{src})$ and $P_r(r^{tgt}|r^{src})$, to map the entities and relations ($e^{src}$ and $r^{src}$) from the source knowledge graph to those ($e^{tgt}$ and $r^{tgt}$) in the target graph, without any labeled data. Towards this goal, we notice that we can align each triplet in the source knowledge graph with one in the target knowledge graph by aligning the head/tail entities and relation respectively. Ideally, the optimal alignment functions would align all the source triplets to some valid triplets (i.e., triplets expressing true facts). Therefore, we can enhance the alignment functions by improving the plausibility of the aligned triplets. With this intuition, we train a triplet discriminator to distinguish between the real triplets in the target knowledge graph and those aligned from the source graph, which provides a reward function to measure the plausibility of a triplet. Meanwhile, the alignment functions are optimized to maximize the reward. The above process naturally forms an adversarial training procedure (Goodfellow et al. (2014)). By alternatively optimizing the alignment functions and the discriminator, the discriminator can consistently enhance the alignment functions.

However, the above approach may suffer from the problem of mode collapse (Salimans et al. (2016)). Specifically, many entities in the source knowledge graph may be aligned to only a few entities in the target knowledge graph. This problem can be addressed if the aggregated posterior entity distribution $\sum_{e^{src}} P_e(e^{tgt}|e^{src})P(e^{src})$ derived by the alignment functions matches the prior entity distribution $P(e^{tgt})$ in the target knowledge graph. Therefore, we match them with another adversarial training framework, which shares similar idea with adversarial auto-encoders (Makhzani et al. (2015)).

The whole framework can also be seamlessly integrated with existing supervised methods, in which we can use a few aligned entities or relations as guidance, yielding a weakly-supervised approach. Our approach can be effectively optimized with stochastic gradient descent, where the gradient for the alignment functions is calculated by the REINFORCE algorithm (Williams (1992)). We conduct extensive experiments on several real-world knowledge graphs. Experimental results prove the effectiveness of our proposed approach in both the weakly-supervised and unsupervised settings.

## 2    RELATED WORK

Our work is related to knowledge graph embedding, that is, embedding knowledge graphs into low-dimensional spaces, in which each entity and relation is represented as a low-dimensional vector (a.k.a., embedding). A variety of knowledge graph embedding approaches have been proposed (Bordes et al. (2013); Wang et al. (2013)), which can effectively preserve the semantic similarities of entities and relations into the learned embeddings. We treat these techniques as tools to learn entity and relation embeddings, which are further used as features for knowledge graph alignment.

In literature, there are also some studies focusing on knowledge graph alignment. Most of them perform alignment by considering contextual features of entities and relations, such as their names (Lacoste-Julien et al. (2013)) or text descriptions (Chen et al. (2018); Wang et al. (2012); Wang et al. (2013)). However, such contextual features are not always available, and therefore these methods cannot generalize to most knowledge graphs. In this paper, we consider the most general case, in which only the triplets in knowledge graphs are used for alignment. The studies most related to ours are Zhu et al. (2017a) and Chen et al. (2017a). Similar to our approach, they treat the entity and relation embeddings as features, and jointly train an alignment model. However, they totally rely on the labeled data (e.g., aligned entities or relations) to train the alignment model, whereas our approach incorporates additional signals by using adversarial training, and therefore achieves better results in the weakly-supervised and unsupervised settings.

More broadly, our work belongs to the family of domain alignment, which aims at mapping data from one domain to data in the other domain. With the success of generative adversarial networks (Goodfellow et al. (2014)), many researchers have been bringing the idea to domain alignment, getting impressive results in many applications, such as image-to-image translation (Zhu et al. (2017b); Zhu et al. (2017c)), word-to-word translation (Conneau et al. (2017)) and text style transfer (Shen et al. (2017)). They typically train a domain discriminator to distinguish between data points from different domains, and then the alignment function is optimized by fooling the discriminator. Our approach shares similar idea, but is designed with some specific intuitions in knowledge graphs.

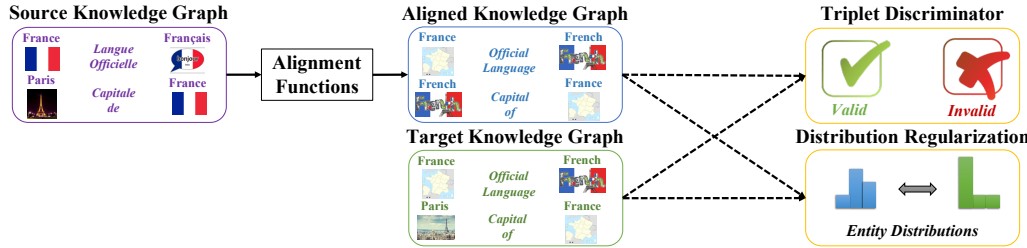

Figure 1: Framework overview. By applying the alignment functions to the triplets in the source knowledge graph, we obtain an aligned knowledge graph. The alignment functions are learned through two GANs. (1) We expect all triplets in the aligned knowledge graph are valid, therefore we train a triplet discriminator to distinguish between valid and invalid triplets, and further use it to facilitate the alignment functions. (2) We also expect the entity distribution in the aligned knowledge graph matches the one in the target knowledge graph, which is achieved with another GAN.

## 3    PROBLEM DEFINITION

**Definition 1** (KNOWLEDGE GRAPH.) *A **knowledge graph** is denoted as $G = (E, R, T)$, where $E$ is a set of entities, $R$ is a set of relations and $T$ is a set of triplets. Each triplet $(h, r, t)$ consists of a head entity $h$, a relation $r$ and a tail entity $t$, meaning that entity $h$ has relation $r$ with entity $t$.*

In practice, the coverage of each individual knowledge graph is usually limited, since it is typically constructed from a single source or language. To construct knowledge graphs with broader coverage, a straightforward way is to integrate multiple knowledge graphs from different sources or languages. However, each knowledge graph uses a unique symbol system to represent entities and relations, which is not compatible with other knowledge graphs. Therefore, a prerequisite for knowledge graph integration is to align entities and relations across different knowledge graphs (a.k.a., knowledge graph alignment). In this paper, we study how to align entities and relations from a source knowledge graph to those in a target knowledge graph, and the problem is formally defined below:

**Definition 2** (KNOWLEDGE GRAPH ALIGNMENT.) *Given a source knowledge graph $G^{src} = (E^{src}, R^{src}, T^{src})$ and a target knowledge graph $G^{tgt} = (E^{tgt}, R^{tgt}, T^{tgt})$, the problem aims at learning an entity alignment function $P_e$ and a relation alignment function $P_r$. Given an entity $e^{src}$ in the source knowledge graph and an entity $e^{tgt}$ in the target knowledge graph, $P_e(e^{tgt}|e^{src})$ gives the probability that $e^{src}$ aligns to $e^{tgt}$. Similarly, for a source relation $r^{src}$ and a target relation $r^{tgt}$, $P_r(r^{tgt}|r^{src})$ gives the probability that $r^{src}$ aligns to $r^{tgt}$.*

## 4    MODEL

In this paper we propose an unsupervised approach to learning the alignment functions, i.e., $P_e(e^{tgt}|e^{src})$ and $P_r(r^{tgt}|r^{src})$, for knowledge graph alignment. To learn them without any supervision, we notice that we can align each triplet in the source knowledge graph with one in the target knowledge graph by aligning the head/tail entities and relation respectively. For an ideal alignment model, all the aligned triplets should be valid ones (i.e., triplets expressing true facts). As a result, we can improve the alignment functions by raising the plausibility of the aligned triplets. With the intuition, our approach trains a triplet discriminator to distinguish between valid triplets and other ones. Then we build a reward function from the discriminator to facilitate the alignment functions. However, using the triplet discriminator alone may cause the problem of mode collapse. More specifically, many entities in the source knowledge graph are aligned to only a few entities in the target knowledge graph. This problem can be addressed if the aggregated posterior distribution of entities derived by the alignment functions matches the prior entity distribution from the target knowledge graph. Therefore, we follow the idea in adversarial auto-encoders (Makhzani et al. (2015)), and leverage another adversarial training framework to regularize the distribution.

The above strategies yield an unsupervised approach. However, in many cases, the structures of the source and target knowledge graphs (e.g., entity and triplet distributions) can be very different, making our unsupervised approach unable to perform effective alignment. In such cases, we can integrate our approach with existing supervised methods, and use a few labeled data as guidance, which further yields a weakly-supervised approach.

Formally, our approach starts by learning entity and relation embeddings with existing knowledge graph embedding techniques, which are denoted as $\{\mathbf{x}_{e^{src}}\}_{e^{src} \in E^{src}}$, $\{\mathbf{x}_{e^{tgt}}\}_{e^{tgt} \in E^{tgt}}$ and $\{\mathbf{x}_{r^{src}}\}_{r^{src} \in R^{src}}$, $\{\mathbf{x}_{r^{tgt}}\}_{r^{tgt} \in R^{tgt}}$. The learned embeddings preserve the semantic correlations of entities and relations, hence we treat them as features and build our alignment functions on top of them. Specifically, we define the probability that a source entity $e^{src}$ or relation $r^{src}$ aligns to a target entity $e^{tgt}$ or relation $r^{tgt}$ as follows:

$$P_e(e^{tgt}|e^{src}) = \frac{\exp(-\gamma||W\mathbf{x}_{e^{src}} - \mathbf{x}_{e^{tgt}}||_2^2)}{Z}, P_r(r^{tgt}|r^{src}) = \frac{\exp(-\gamma||W\mathbf{x}_{r^{src}} - \mathbf{x}_{r^{tgt}}||_2^2)}{Z}, \quad (1)$$

where $\gamma$ is a temperature parameter, $Z$ is a normalization term. $W$ is a linear projection matrix, which maps an embedding in the source knowledge graph (e.g., $\mathbf{x}_{e^{src}}$) to one in the target graph (e.g., $W\mathbf{x}_{e^{src}}$), so that we can perform alignment by calculating the distance between the mapped source embeddings (e.g., $W\mathbf{x}_{e^{src}}$) and the embeddings in the target graph (e.g., $\mathbf{x}_{e^{tgt}}$). Note that $W$ is the only parameter to be learned, and it is shared across the entity and relation alignment functions. We also try independent projection matrices or nonlinear projections, but get inferior results.

In the following chapters, we first briefly introduce the method for learning entity and relation embeddings (Section 4.1). Then, we introduce how we leverage the triplet discriminator (Section 4.2) and the regularization mechanism (Section 4.3) to facilitate training the alignment functions. Afterwards, we introduce a simple supervised method as an example, to illustrate how to incorporate labeled data (Section 4.4). Finally, we introduce our optimization algorithm (Section 4.5).

## 4.1 Entity and Relation Embedding Learning

In this paper, we leverage the TransE algorithm (Bordes et al. (2013)) for entity and relation embedding learning, due to its simplicity and effectiveness in a wide range of datasets. In general, we can also use other knowledge graph embedding algorithms as well.

Given a triplet $t = (e_h, r, e_t)$, TransE defines its score as follows:

$$\text{score}(t) = -||\mathbf{x}_{e_h} + \mathbf{x}_r - \mathbf{x}_{e_t}||_2. \quad (2)$$

Then the model is trained by maximizing the margin between the scores of real triplets and random triplets, and the objective function is given below:

$$O_{TransE} = \mathbb{E}_{t \in T, t' \in T'} \min(\text{score}(t) - \text{score}(t') - m, 0), \quad (3)$$

where $T$ is the set of real triplets in the knowledge graph, $T'$ is the set of random triplets, and $m$ is a parameter controlling the margin.

## 4.2 Learning with Triplet Discriminators

By defining the alignment functions for entity and relation (Eqn. 1), we are able to align each triplet in the source knowledge graph to the target knowledge graph by aligning the entities and relation respectively. An ideal alignment function would align all the source triplets to some valid triplets. Therefore, we can enhance the alignment functions by raising the plausibility of the aligned triplets.

Formally, for each triplet $(e_h^{src}, r^{src}, e_t^{src})$ in the source knowledge graph, we can sample an aligned triplet $(e_h^{tgt}, r^{tgt}, e_t^{tgt})$ by mapping the entities and relation to the target knowledge graph with the alignment functions (Eqn. 1). The process is defined below:

$$(e_h^{tgt}, r^{tgt}, e_t^{tgt}) \sim A(e_h^{src}, r^{src}, e_t^{src}) := e_h^{tgt} \sim P_e(\cdot|e_h^{src}), r^{tgt} \sim P_r(\cdot|r^{src}), e_t^{tgt} \sim P_e(\cdot|e_t^{src}). \quad (4)$$

Ideally, we would wish that all the aligned triplets are valid ones. Towards this goal, we train a triplet discriminator to distinguish between valid triplets and other ones. Then the discriminator is used to define different reward functions for guiding the alignment functions. In our approach, we train the discriminator by treating the real triplets in knowledge graphs as positive examples, and the aligned triplets generated by our approach as negative examples. Following existing studies (Goodfellow et al. (2014)), we define the objective function below:

$$O_{D_t} = \mathbb{E}_{t^{tgt} \sim T^{tgt}}[\log D_t(t^{tgt})] + \mathbb{E}_{t^{src} \sim T^{src}, t \sim A(t^{src})}[\log(1 - D_t(t)], \quad (5)$$

where $t \sim A(t^{src})$ is a triplet aligned from $t^{src}$ and $A$ is defined in Eqn. 4. $D_t$ is the triplet discriminator, which concatenates the embeddings of the head/tail entities and relation in a triplet $t$, and further predicts the probability that $t$ is a valid triplet.

Based on the discriminator, we can construct a scalar-to-scalar reward function $R$ to measure the plausibility of a triplet. Then the alignment functions can be trained by maximizing the reward:

$$O_T = \mathbb{E}_{t^{src} \sim T^{src}, t \sim A(t^{src})}[R(D_t(t))]. \tag{6}$$

There are several ways to define the reward function $R$, which essentially yields different adversarial training frameworks. For example, Goodfellow et al. (2014) and Ho & Ermon (2016) treat $R(x) = \log x$ as the reward function. Finn et al. (2016) uses $R(x) = \log \frac{x}{1-x}$. Che et al. (2017) considers $R(x) = \frac{x}{1-x}$. Besides, we may also leverage $R(x) = x$, which is the first-order Taylor's expansion of $-\log(1-x)$ at $x = 1$ and has a limited range when $x \in (0, 1)$. All different reward functions have the same optimal solution, i.e, the derived distribution of the aligned triplets matching the real triplet distribution in the target knowledge graph. In practice, these reward functions may have different variance, and we empirically compare them in the experiments (Table 5).

During optimization, the gradient with respect to the alignment functions cannot be calculated directly, as the triplets sampled from the alignment functions are discrete variables. Therefore, we leverage the REINFORCE algorithm (Williams (1992)), which calculates the gradient as follows:

$$\nabla_W O_T = \mathbb{E}_{t^{src} \sim T^{src}, t \sim A(t^{src})}[R(D_t(t))\nabla_W \log P(t|t^{src})], \tag{7}$$

where $P(t|t^{src}) = P_e(e_h|e_h^{src})P_r(r|r^{src})P_e(e_t|e_t^{src})$ with $t = (e_h, r, e_t)$, $t^{src} = (e_h^{src}, r^{src}, e_t^{src})$.

### 4.3 Constraining the Aggregated Posterior Distribution

Although the triplet discriminator provides effective reward to the alignment functions, many entities in the source knowledge graph can be aligned to only a few entities in the target knowledge graph. Such problems can be solved by constraining the aggregated posterior entity distribution derived by the alignment functions to match the prior entity distribution in the target knowledge graph.

Formally, the aggregated posterior distribution of entities is given below:

$$\hat{P}(e^{tgt}) = \sum_{e^{src}} P_e(e^{tgt}|e^{src})P(e^{src}), \tag{8}$$

where $P(e^{src})$ is the entity distribution in the source knowledge graph. We expect this distribution to match the prior distribution $P(e^{tgt})$, which is the entity distribution in the target knowledge graph.

Following Makhzani et al. (2015), we regularize the distribution with another adversarial training framework (Goodfellow et al. (2014)). During training, an entity discriminator $D_e$ is learned to distinguish between the posterior and prior distributions using the following objective function:

$$O_{D_e} = \mathbb{E}_{e^{tgt} \sim P(e^{tgt})}[\log D_e(e^{tgt})] + \mathbb{E}_{e \sim \hat{P}(e^{tgt})}[\log(1 - D_e(e)], \tag{9}$$

where $D_e$ takes the embedding of an entity as features to predict the probability that the entity is sampled from prior distribution $P(e^{tgt})$. To enforce the posterior distribution to match the prior distribution, the entity alignment function is trained to fool the discriminator by maximizing the following objective:

$$O_E = \mathbb{E}_{e \sim \hat{P}(e^{tgt})}[R(D_e(e))], \tag{10}$$

where $R$ is the same reward function as used in the triplet discriminator (Eqn. 6), and the gradient for the alignment functions can be similarly calculated with the REINFORCE algorithm.

### 4.4 Weakly-supervised Learning

The above sections introduce an unsupervised approach to knowledge graph alignment. In many cases, the source and target knowledge graphs may have very different structures (e.g., entity or triplet distributions), making our approach fail to perform effective alignment. In these cases, we can integrate our approach with any supervised methods, and leverage a few labeled data (e.g., aligned entity or relation pairs) as guidance, which yields a weakly-supervised approach. In this section, we introduce a simple yet effective method to show how to utilize the labeled data.

Suppose we are given some aligned entity pairs, and the aligned relation pairs can be handled in a similar way. We define our objective function as follows:

$$O_L = \mathbb{E}_{(e^{src}, e^{tgt}) \in S} \log P_e(e^{tgt}|e^{src}) - \lambda H(P_e(\mathbf{e}^{tgt}|\mathbf{e}^{src})), \tag{11}$$

where $S$ is the set of aligned entity pairs, $\mathbf{e}^{src}$ and $\mathbf{e}^{tgt}$ are random variables of entities in the source and target knowledge graphs, $H$ is the entropy of a distribution. The first term corresponds to a softmax classifier, which aims at maximizing the probability of aligning a source entity to the ground-truth target entity. The second term minimizes the entropy of the probability distribution calculated by the alignment function, which encourages the alignment function to make confident predictions. Such an entropy minimization strategy is used in many semi-supervised learning studies (Grandvalet & Bengio (2005)).

### 4.5 OPTIMIZATION

We leverage the stochastic gradient descent algorithm for optimization. In practice, we find that first pre-training the alignment functions with the given labeled data (Eqn. 11), then fine-tuning them with the triplet discriminator (Eqn. 6) and the regularization mechanism (Eqn. 8) leads to better performance, compared with jointly training all of them (Table 6). Consequently, we adopt the pre-training and fine-tuning framework for optimization, which is summarized in Alg. 1.

---

**Algorithm 1** Optimization algorithm.

---

**Input:** Two knowledge graphs $G^{src}$ and $G^{tgt}$, some aligned entity/relation pairs $S$ (optional).
**Output:** The learned alignment functions $P_e$ and $P_r$.
 1: Pre-train entity and relation embeddings by optimizing Eqn. 3.
 2: Pre-train the alignment functions by optimizing Eqn. 11 or using any other supervised methods.
 3: **while** not converge **do**
 4:     Update the triplet discriminator $D_t$ and the alignment functions with Eqn. 5 6.
 5:     Update the entity discriminator $D_e$ and the alignment functions with Eqn. 9 10.
 6: **end while**

---

## 5 EXPERIMENT

### 5.1 EXPERIMENT SETUP

In experiment, we use four datasets for evaluation. In FB15k-1 and FB15k-2, the knowledge graphs have very different triplets, which can be seen as constructed from different sources; in WK15k(en-fr) and WK15k(en-de), the knowledge graphs are from different languages. The statistics are summarized in Table 1. Following existing studies (Zhu et al. (2017a); Chen et al. (2017a)), we consider the task of entity alignment, and three different settings are considered, including supervised, weakly-supervised and unsupervised settings. Hit@1 and Hit@10 are reported.

Table 1: Statistics of the Datasets.

| Dataset | FB15k-1 | | FB15k-2 | | WK15k(en-fr) | | WK15k(en-de) | |
|---|---|---|---|---|---|---|---|---|
| | src | tgt | src | tgt | en | fr | en | de |
| #Entities | 14,951 | 14,951 | 14,951 | 14,951 | 15,169 | 15,392 | 15,125 | 14,602 |
| #Relations | 1,345 | 1,345 | 1,345 | 1,345 | 2,217 | 2,416 | 1,833 | 594 |
| #Triplets | 444,159 | 444,160 | 325,717 | 325,717 | 203,226 | 170,441 | 210,611 | 145,567 |
| #Training Pairs (full) | 5,000 | | 500 | | 3,874 (en2fr) | 3,856 (fr2en) | 7,853 (en2de) | 5,606 (de2en) |
| #Training Pairs (weak) | 50 | | 50 | | 500 (en2fr) | 500 (fr2en) | 50 (en2de) | 50 (de2en) |
| #Test Pairs | 9,951 | | 14,451 | | 3,731 (en2fr) | 3,809 (fr2en) | 1,832 (en2de) | 1,609 (de2en) |

**Datasets**

- **FB15k-1, FB15k-2**: Following Zhu et al. (2017a), we construct two datasets from the FB15k dataset (Bordes et al. (2013)). In FB15k-1, the two knowledge graphs share 50% triplets, and in FB15k-2 10% triplets are shared. According to the study, we use 5000 and 500 aligned entity pairs as labeled data in FB15k-1 and FB15k-2 respectively, and the rest for evaluation.
- **WK15k(en-fr)**: A bi-lingual (English and French) dataset in Chen et al. (2017a). Some aligned triplets are provided as labeled data, and some aligned entity pairs as test data. The labeled data and test data have some overlaps, so we delete the overlapped pairs from labeled data.
- **WK15k(en-de)**: A bi-lingual (English and German) dataset used in Chen et al. (2017a). The dataset is similar to WK15k(en-fr), so we perform preprocessing in the same way.

**Compared Algorithms**

(1) **iTransE** (Zhu et al. (2017a)): A supervised method for knowledge graph alignment. (2) **MLKGA** (Chen et al. (2017a)): Another supervised method for multi-lingual knowledge graph alignment. (3) **Procrustes** (Artetxe et al. (2017)): A supervised method for word translation, which learns the translation in a bootstrapping way. We apply the method on the pre-trained entity and relation embeddings to perform knowledge graph alignment. (4) **UWT** (Conneau et al. (2017)): An unsupervised word translation method, which leverages adversarial training and a refinement strategy. We apply the method to the entity and relation embeddings to perform alignment. (5) **KAGAN-sup**: The supervised method introduced in Section 4.4, which is simple but effective, and can be easily integrated with our unsupervised approach. (6) **KAGAN**: Our proposed Knowledge-graph Alignment GAN, which leverages the labeled data for pre-training, and then fine-tunes the model with the triplet discriminator and the regularization mechanism. (7) **KAGAN-t**: A variant with only the triplet GAN, which first performs pre-training with the labeled data, and then performs fine-tuning with the triplet discriminator. (8) **KAGAN-e**: A variant with only the entity GAN, which first pre-trains with the labeled data, and then fine-tunes with the regularization mechanism.

### Parameter Settings

The dimension of entity and relation embeddings is set as 512 for all compared methods. For our proposed approach, $\lambda$ is set as 1, the temperature parameter $\gamma$ is set as 1, and the reward function is set as $x$ by default. SGD is used for optimization. The learning rate is set as 0.1 for pre-training and 0.001 for fine-tuning. 10% labeled pairs are treated as the validation set. The training process is terminated if the performance on the validation set drops. For the compared methods, the parameters are chosen according to the performance on the validation set.

### 5.2 QUANTITATIVE RESULTS

Table 2: Quantitative Results of Entity Alignment.

| Setting | Algorithm | FB15k-1 | | FB15k-2 | | WK15k fr2en | | WK15k en2fr | | WK15k de2en | | WK15k en2de | |
|---|---|---|---|---|---|---|---|---|---|---|---|---|---|
| | | P@1 | P@10 | P@1 | P@10 | P@1 | P@10 | P@1 | P@10 | P@1 | P@10 | P@1 | P@10 |
| Supervised | iTransE | 64.47 | 80.74 | 9.64 | 29.08 | 0.63 | 8.43 | 0.43 | 9.33 | 3.79 | 8.89 | 5.68 | 11.30 |
| | MLKGA | 78.73 | 90.50 | 53.33 | 78.41 | 17.83 | 41.80 | 17.53 | 41.97 | 42.76 | 57.55 | 32.86 | 50.98 |
| | Procrustes | 82.36 | 92.13 | 72.08 | 87.15 | 21.82 | 44.21 | 20.72 | 43.74 | 45.62 | 58.11 | 34.28 | 51.15 |
| | KAGAN-sup | 82.27 | 92.01 | 71.86 | 87.30 | 22.39 | 44.11 | 22.46 | 43.50 | 44.81 | 58.17 | 34.12 | 50.93 |
| | KAGAN | 84.93 | 93.73 | 74.04 | 88.21 | 25.15 | 45.16 | 25.03 | 44.79 | 48.10 | 58.79 | 39.47 | 52.57 |
| Unsupervised | UWT | 79.33 | 91.48 | 70.03 | 86.86 | 0.66 | 2.97 | 0.03 | 0.46 | 0.44 | 1.55 | 0.55 | 3.06 |
| | KAGAN | 84.12 | 93.14 | 73.29 | 87.89 | 1.73 | 3.33 | 0.27 | 1.29 | 1.31 | 2.18 | 2.07 | 4.15 |
| Weakly-supervised | Procrustes | 81.97 | 92.04 | 71.64 | 87.03 | 17.85 | 39.62 | 17.72 | 40.77 | 40.71 | 53.82 | 30.57 | 45.80 |
| | KAGAN | 84.60 | 93.51 | 73.75 | 88.10 | 20.82 | 42.16 | 20.69 | 43.50 | 45.68 | 58.11 | 34.83 | 49.95 |

Table 3: Ablation Study (Full Supervision).

| Algorithm | FB15k-2 | | WK15k fr2en | | WK15k de2en | |
|---|---|---|---|---|---|---|
| | P@1 | P@10 | P@1 | P@10 | P@1 | P@10 |
| KAGAN-sup | 71.86 | 87.30 | 22.39 | 44.11 | 44.81 | 58.17 |
| KAGAN-e | 72.82 | **88.23** | 23.92 | 44.74 | 47.23 | 58.67 |
| KAGAN-t | 73.97 | 88.06 | 24.65 | 44.84 | 47.92 | **59.04** |
| KAGAN | **74.04** | 88.21 | **25.15** | **45.16** | **48.10** | 58.79 |

Table 4: Ablation Study (Weak/No Supervision).

| Algorithm | FB15k-2 | | WK15k fr2en | | WK15k de2en | |
|---|---|---|---|---|---|---|
| | P@1 | P@10 | P@1 | P@10 | P@1 | P@10 |
| KAGAN-sup | 0 | 0 | 17.35 | 39.12 | 40.27 | 53.39 |
| KAGAN-e | 70.51 | 86.26 | 18.93 | 40.14 | 42.20 | 55.38 |
| KAGAN-t | 0.09 | 1.02 | 19.74 | 41.45 | 44.25 | 57.55 |
| KAGAN | **73.29** | **87.89** | **20.82** | **42.16** | **45.68** | **58.11** |

The main results are presented in Table 2. In the supervised setting, our approach significantly outperforms all the compared methods. On the FB15k-1 and FB15k-2 datasets, without using any labeled data, our approach already achieves close results as in the supervised setting. On the WK15k datasets under the weakly-supervised setting, our performance is comparable or even superior to the performance of other methods in the supervised setting, but with much fewer labeled data (about 13% in WK15k(en-fr) and 1% in WK15k(en-de)). Overall, our approach is quite effective in the weakly-supervised and unsupervised settings, outperforming all the baseline approaches.

To understand the effect of each part in our approach, we further conduct some ablation studies. Table 3 presents the results in the supervised setting. Both the triplet discriminator (KAGAN-t) and the regularization mechanism (KAGAN-e) improves the pre-trained alignment models (KAGAN-pre). Combining them (KAGAN) leads to even better results. Table 4 gives the results in the unsupervised (FB15k-2) and weakly-supervised (WK15k-fr2en, WK15k-de2en) settings. On the FB15k-2 dataset, using the regularization mechanism alone (KAGAN-e) already achieves impressive results. This is because the source and target knowledge graphs in FB15k-2 share similar structures, and our regularization mechanism can effectively leverage such similarity for alignment. However, the

performance of using only the triplet discriminator (KAGAN-t) is very poor, which is caused by the problem of mode collapse. The problem is effectively solved by integrating the approach with the regularization mechanism (KAGAN), which achieves the best results in all cases.

Table 5: Comparison of Reward Functions.

| Reward Function | WK15k fr2en | WK15k de2en |
|---|---|---|
| 0 (w/o reward) | 22.39 | 44.81 |
| $\log x$ | 23.92 | 47.36 |
| $\log \frac{x}{1-x}$ | 24.05 | 47.30 |
| $\frac{x}{1-x}$ | 25.07 | 47.98 |
| $x$ | **25.15** | **48.10** |

Table 6: Comparison of Optimization Methods.

| Method | WK15k fr2en | WK15k de2en |
|---|---|---|
| Joint train | 23.47 | 45.99 |
| Pre-train and fine-tune | **25.15** | **48.10** |
| w embedding tuning | 24.21 | 47.42 |
| w/o embedding tuning | **25.15** | **48.10** |

## 5.3 PERFORMANCE ANALYSIS

**Comparison of the reward functions.** In our approach, we can choose different reward functions, leading to different adversarial training frameworks. These frameworks have the same optimal solutions, but with different variance. In this section we compare them on the WK15k datasets, and the results of Hit@1 are presented in Table 5. We notice that all reward functions lead to significant improvement compared with using no reward. Among them, $\frac{x}{1-x}$ and $x$ obtain the best results.

**Comparison of the optimization methods.** During training, our approach fixes the entity/relation embeddings, and uses a pre-training and fine-tuning framework for optimization. In this section, we compare the framework with some variants, and the results of Hit@1 are presented in Table 6. We see that our framework (pre-training and fine-tuning) outperforms the joint training framework. Besides, fine-tuning entity/relation embeddings yields worse results than fixing them during training.

**Case study.** In this section, we present some visualization results to intuitively show the effect of the triplet discriminator and regularization mechanism in our approach. We consider the unsupervised setting on the FB15k-2 dataset, and leverage the PCA algorithm to visualize certain embeddings. Figure 2 compares the entity embeddings obtained with and without the regularization mechanism, where red is for the mapped source entity embeddings ($W\mathbf{x}_{e^{src}}$), and green for the target embeddings ($\mathbf{x}_{e^{tgt}}$). We see that without the mechanism, many entities from the source knowledge graph are mapped to a small region (the red region), leading to the problem of mode collapse. The problem is addressed with the regularization mechanism. Figure 3 compares the triplet embeddings obtained with and without the triplet discriminator, where the triplet embedding is obtained by concatenating the entity and relation embeddings in a triplet. Red color is for triplets aligned from the source knowledge graph, and green is for triplets in the target graph. Without the triplet discriminator, the aligned triplets look quite different from the real ones (under different distributions). With the triplet discriminator, the aligned triplets look like the real ones (under similar distributions).

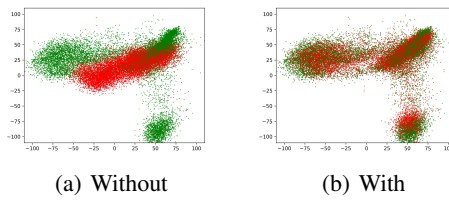

(a) Without      (b) With

Figure 2: The Regularization Mechanism.

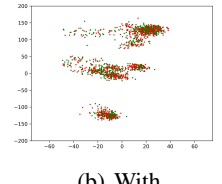

(a) Without      (b) With

Figure 3: The Triplet Discriminator.

## 6 CONCLUSION

This paper studied knowledge graph alignment. We proposed an unsupervised approach based on the adversarial training framework, which is able to align entities and relations from a source knowledge graph to those in a target knowledge graph. In practice, our approach can be seamlessly integrated with existing supervised methods, which enables us to leverage a few labeled data as guidance, leading to a weakly-supervised approach. Experimental results on several real-world datasets proved the effectiveness of our approach in both the unsupervised and weakly-supervised settings. In the future, we plan to learn alignment functions from two directions (source to target and target to source) to further improve the results, which is similar to CycleGAN (Zhu et al. (2017b)).

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
