# OpenReview forum: "Weakly-supervised Knowledge Graph Alignment with Adversarial Learning"
_ICLR.cc/2019/Conference_

### Official Review · AnonReviewer1 · 2018-11-02
**Interesting paper, but does it really need RL?**

**Rating:** 5
**Confidence:** 3

**Review:**

This paper focuses on the alignment of different Knowledge Graphs (KGs) obtained from multiple sources and languages - this task is similar to the Link Prediction setting, but the objective is learning a mapping from the entities (or triples) in one Knowledge Graph to another. In particular, the paper focuses on the setting where the number of available training alignments is small.

The model and training processes are slightly convoluted:
- Given a triple in the KG A, the model samples a candidate aligned triple from a KG B, from a (learned) alignment distribution.
- The objective is a GAN loss where the discriminator needs to distinguish between real and generated alignments.
The objective is non-differentiable (due to the sampling step), and it's thus trained via policy gradients.

Question: to me it looks like the whole process could be significantly simpler, and end-to-end differentiable. For instance, the loss may be the discrepancy between the alignment distribution and the training alignments. As a consequence, the whole procedure would be significantly more stable; there would be no need of sampling; or tricks for reducing the variance of the gradient estimates. What would happen with such a model? Would it be on par with the proposed one?

The final model seems to be better than the considered baselines.

---

### Official Review · AnonReviewer2 · 2018-11-03
**Interesting approach  for aligning partially heterogeneous knowledge graphs**

**Rating:** 5
**Confidence:** 4

**Review:**

<Summary>
Authors propose a new approach for aligning partially heterogeneous knowledge graphs, which allows for unsupervised / semi-supervised alignment via GANs that serve as: (1) triplet discriminator and (2) entity distribution matching.

The paper reports big improvement over their baseline approaches especially for unsupervised and weakly-supervised settings. The experiment is performed for various combinations of several small KG datasets (both simulated - artificial division of a KB - and real multi-lingual KG triple datasets). The authors also report ablation studies over multiple optimization algorithms as well as reward functions,

<Comments>
- As far as I know, the idea of applying GAN for triplet discrimination has not been tried before, and it is a novel contribution of the authors. The prior literature has focused primarily on distribution matching / regularization only, often via MMR, autoencoders, GAN training, etc.

- While the idea of applying adversarial approaches is new in the context of KG alignment, the technical novelty of each component is limited, and borrowed directly from existing literature with minor modifications.

- It would be interesting to report results at varying controlled conditions (e.g. as a function of % of label overlaps, etc.) to better study the empirical effects of each training component, etc.

- Experiments are only conducted on smaller 15K subsets of FB KG datasets. It would be interesting to run experiments on larger subsets and see if the proposed approaches can scale.

- The alignment is applied only on a simple TransE algorithm.  While the author’s formulation is flexible and shouldn’t restrict application of other state-of-the-art graph embedding methods, it is not clear if one will see the same improvement.

---

### Official Review · AnonReviewer3 · 2018-11-04
**The biggest strength of the paper is the novelty of the proposed method but we are not happy with the experimental evaluation.**

**Rating:** 5
**Confidence:** 3

**Review:**

The authors propose KAGAN, a novel method for knowledge graph (KG) alignments using GANs. In contrast to most other methods, KAGAN does not rely on a supervised setting where a set of already aligned triples is used as seed. In addition, the authors propose modifications such that their method can also integrate information about aligned triples.

In my opinion the biggest strength of the paper is the novelty of the proposed method. The standard framework of GANs is adjusted elegantly to the KG alignment setting. Further, the basic approach as well as the proposed modification to deal with practical issues such as mode collapse are well motivated and comprehensible. The ability to perform well in the unsupervised/weakly supervised setting is another plus. Up to my knowledge, this paper constitutes the first approach using GANs for the KG alignment task. However, there are several methods that use GANs directly to produce KG embeddings (e.g. KBGAN). These methods are related to KAGAN since they employ TransE as an underlying embedding method. Therefore, I strongly believe the authors should include [1] and [2] in their references and discuss these methods briefly.

The authors conduct experiments on benchmark datasets for the KG alignment tasks. Thereby, they aim to show that their KAGAN outperforms other state of the art methods. However, I have one major concern: In the subsection 'Parameter Settings' the authors state without further explanation that they set the embedding size to 512 for all compared methods. I do not understand the rationale behind this decision. In particular, since the embedding sizes in the original publications (e.g., in Chen et al. (2017a)) are very different. For KG embedding methods the dimensionality of the embeddings is probably the most important hyperparameter. In my opinion, picking a global value for all methods instead of tuning it for each method individually does not guarantee a fair comparison.

On a side note, the authors might also consider the mean rank as performance measure which is frequently employed in the KG alignment setting.

Overall, the paper is well written and organized. A minor point: I would get rid of the formulas in the introduction; in particular, since the notation is not introduced at this point.

While the novelty and good structure of the paper are reasons to accept it, I have doubts concerning the soundness of the results due to the experimental setup.

Reasons to accept the paper
- novelty
- works in an unsupervised setting
- well written/structured

Reasons to reject
-  Doubts concerning the experimental setting
- (Minor) related work is not complete
- (Minor) not all common performance measures are reported


----------------------------

[1] Wang, Peifeng, Shuangyin Li, and Rong Pan. "Incorporating GAN for Negative Sampling in Knowledge Representation Learning." (2018).
[2] Cai, Liwei, and William Yang Wang. "KBGAN: Adversarial Learning for Knowledge Graph Embeddings." Proceedings of the 2018 Conference of the North American Chapter of the Association for Computational Linguistics: Human Language Technologies, Volume 1 (Long Papers). Vol. 1. 2018.

---

### Public Comment · (anonymous) · 2018-10-29
**Related work**

An interesting work! However, it seems that this paper omits BootEA[1], one of  the SOTA semi-supervised models for entity alignment. I would suggest the authors to cite it and provide a discussion or comparison.

[1] Zequn Sun, Wei Hu, Qingheng Zhang, Yuzhong Qu: Bootstrapping Entity Alignment with Knowledge Graph Embedding. IJCAI 2018: 4396-4402

---

> ### Author Response · Authors · 2018-10-29
> **Response**
>
> Thanks for pointing it out. We will add the paper in the revised version.

---

### Meta-Review · Area_Chair1 · 2018-12-17
**Limited evaluation**

**Confidence:** 4
**Recommendation:** Reject

**Metareview:**

This paper considers an important problem of aligning two knowledge graphs (the entities and relations therein). The reviewers found the use of adversarial training quite novel and appropriate for the the task, especially as it works in the unsupervised setting as well. The reviewers were also impressed that the proposed work outperforms existing approaches in terms of the accuracy of the alignment.

The following potential weaknesses were raised by the reviewers and the AC: (1) Reviewer 3 brings up the fact that the hyperaparameters were set different from the original publications of the baselines, and thus are not convinced of the soundness of the results, (2) Reviewer 2 notes that the evaluation is limited, and more variations should be considered, such as varying the overlap, taking larger subsets of knowledge graphs, and going beyond TranE as the choice for embedding, and (3) Reviewer 3 notes that a simpler baseline based on alignment discrepancy should be  considered, which would alleviate the need for RL based training.

Although the reviewers raised very different concerns with the paper, none of them were addressed in a response or revision, and thus they agree that the paper should be rejected.